# Metal-organic framework derived crystalline nanocarbon for Fenton-like reaction

Tingting Lian [1], Li Xu[2], Diana Piankova[1], Jin-Lin Yang[3], Nadezda V. Tarakina [1], Yang Wang [1,2] ✉ & Markus Antonietti [1]

Nanoporous carbons with tailorable nanoscale texture and long-range ordered structure are promising candidates for energy, environmental and catalytic applications, while the current synthetic methods do not allow elaborate control of local structure. Here we report a salt-assisted strategy to obtain crystalline nanocarbon from direct carbonization of metal-organic frameworks (MOFs). The crystalline product maintains a highly ordered two-dimensional (2D) stacking mode and substantially differs from the traditional weakly ordered patterns of nanoporous carbons upon high-temperature pyrolysis. The MOF-derived crystalline nanocarbon (MCC) comes with a high level of nitrogen and oxygen terminating the 2D layers and shows an impressive performance as a carbocatalyst in Fenton-like reaction for water purification. The successful preparation of MCC illustrates the possibility to discover other crystalline heteroatom-doped carbon phases starting from correctly designed organic precursors and appropriate templating reactions.

Crystalline nanomaterials with long-range ordered structures and uniformly distributed active sites are powerful in advancing catalytic activity and selectivity[1-3]. This has been exemplified by recent progress in zeolites and metal-organic frameworks (MOFs), which are emerging candidates and increasingly prevalent within a broad energy- and environmental-related scope, due to the combined merits of topological diversity, high surface area, permanent porosity and functionality tunability[4-7]. Such highly ordered materials with well-defined nanostructure and pore systems allow molecule-level design to afford diverse catalytic functionalities with substrate specificity, while concomitantly ensuring the diffusion and selection of guest molecules in the pore system and at interfaces[8,9]. Further deployment of these systems is however restricted by the limited range of active sites and low stability, an inherent downturn originating from the relatively weak metal-ligand coordination needed for the organization of structure[10]. In this context, derivatives of MOFs, specifically, MOF-derived nanoporous carbons (NPCs), are proposed to address the stability issue and impart the thermally treated products with enhanced electron transfer kinetics[11-13]. NPC materials are recognized as an ideal platform to tackle questions of energy change, environmental pollution, biosafety and beyond[14-16].

The first-generation MOF-derived NPCs were constructed by direct carbonization of MOF precursors[11,17], leaving the resultant powders at best under preservation of texture, but otherwise present only a featureless nanostructure (Fig. 1). Further efforts have been devoted to preserving the morphologies of pristine MOFs using external templates (e.g. $SiO_2$[18], $TiO_2$[19], $ZnO$[20]), but this comes with elaborated procedures to remove the post-synthetic matrices, and local structure was still rather disordered. In comparison, self-templating methods have the potential to proceed in a more feasible way, while only a handful of MOFs with sufficient stability are available, mostly limited to zeolitic imidazolate framework (ZIF) series[12,21,22]. Alternative strategies incorporated with polymers or surfactants were also proposed to regulate the morphology of NPCs, where the van-der-Waals interaction between MOFs and modifiers is assumed to prevent the collapse of nanostructures. Unfortunately, these structure-directing amphiphiles convert into skin layers with different compositions which then decrease accessible surface area and block the active sites. Note that the state-of-the-art NPCs, that is, the second-

[1]Department of Colloid Chemistry, Max Planck Institute of Colloids and Interfaces, 14476 Potsdam, Germany. [2]Department of Environmental Science and Engineering, University of Science and Technology of China, 230026 Hefei, China. [3]School of Physical and Mathematical Sciences, Nanyang Technological University, 637371 Singapore, Singapore. ✉e-mail: ywangese@ustc.edu.cn

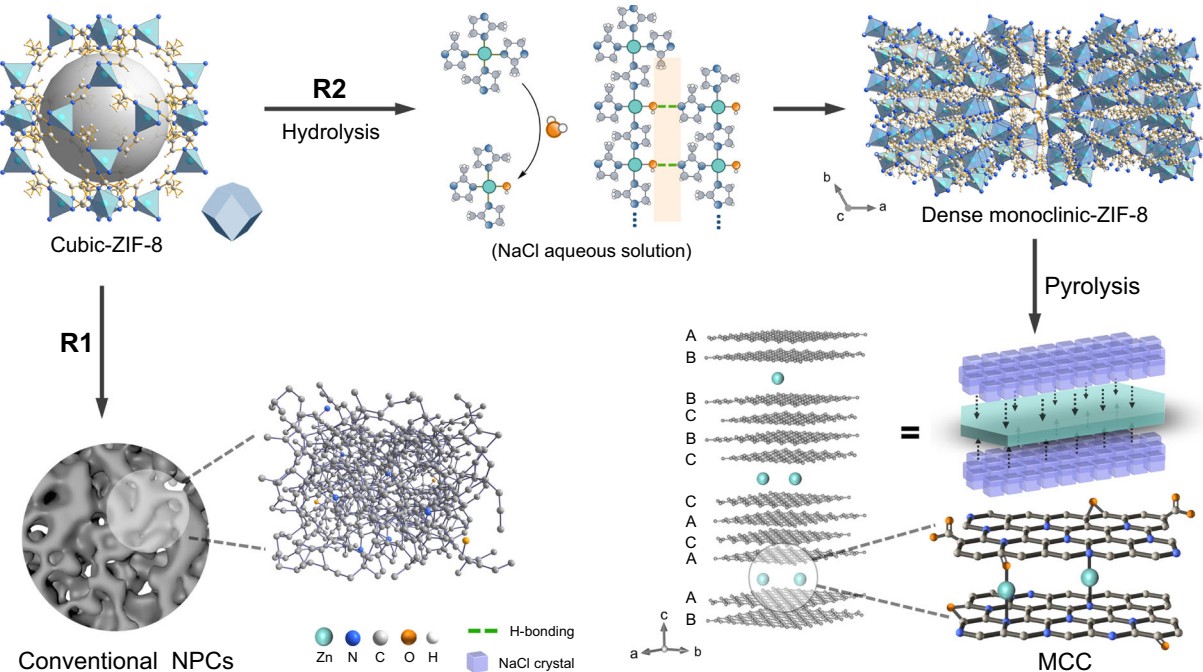

**Fig. 1 | Synthetic strategy of MOF-derived carbons.** R1: default direct carbonization route leading to disordered NPCs, R2: optimized salt-assisted route with a combination of salt-induced recrystallization, hybrid formation and thermal condensation for the preparation of MCC. The orange region denotes the OH--N hydrogen bonding.

generation MOF-derived NPCs, are all characterized by a disordered carbon nanostructure, with the absence of a periodic and well-defined framework (Supplementary Table 1). Although NPCs provide access for reactants or adsorbates under improved mass and electron transfer, the disordered local carbon structure still creates a polytype of active sites as there is no local control of the active centers[23,24]. It is still a riddle how to establish crystalline local order in NPCs with surface functionality relevant to improved catalytic performances.

In the high-temperature processing of MOF materials, it is well acknowledged that the organic ligands can affect the nucleation and growth of metal oxides, which in turn impose steric constraints on the condensation of the organic linkers[6]. This synergetic effect between inorganic and organic condensation processes sets a base for potential topotactic phase transformation, if such interactions are finely regulated before and during carbonization of MOFs. In this regard, the salt-assisted synthesis provides a protective and homogeneous reaction environment and offers an additional tool to avoid the collapse of the precursor structure and the aggregation of final products[25–29]. Specifically, high-melting-point salts that survive at high temperatures can, in principle, help configuring materials with high crystallinity. Unlike other strongly acidic or basic salt species (e.g. $ZnCl_2$, LiCl)[30], the relatively mild NaCl may fit the purpose properly, which is in addition cheap and easy to remove (Supplementary Note 1).

Herein, we present the synthesis of a highly organized, crystalline NPCs via a facile NaCl-assisted strategy using monoclinic ZIF-8 sealed with NaCl crystals, followed by thermal treatment (Fig. 1). The widely reported cubic ZIF-8 reorganizes in aqueous NaCl solution into stacked sheets of the monoclinic phase, wherein NaCl serves as structure-directing surface modifier. NaCl at higher temperatures is also identified as a protective ex-template that controls the synergistic inorganic and organic phase transformations, while a type of epitaxy favors structural optimization and regularity of the heteroatoms terminating the carbon subphase. These self-processes drive the preparation of a highly ordered, MOF-derived, crystalline nanocarbon (MCC), to be carefully distinguished from all disordered NPCs of previous reports.

This MCC with its organized oxygen (O) and nitrogen (N) heterotermination is then used as a carbocatalyst to tackle a critical environmental issue, *i.e.*, remediation of polluted water. We observe fast kinetics in the advanced oxidation process via a Fenton-like reaction, in which MCC catalytically removes even rather inert organic pollutants from water at the seconds scale, ranking it among the best carbocatalysts. The combined findings in this work demonstrate the possibility of configuring crystalline nanocarbons even at high pyrolysis temperatures, and prove in general relevant catalytic activity of carbon-based organic solids, which may qualify their uses in energy conversion, environmental remediation, adsorption, and separation.

## Results

### Preparation and characterizations of the catalyst

Previous experiments already evidenced the salt-templating effect of NaCl for the preparation of NPCs[31–33] and carbon nitrides[34,35]. The presence of salt reduces the volatility of organic intermediates during the carbonization and stimulates optimization of local structure by improving bond exchange, both driving higher mass yields. It is however noticed that when NaCl was mixed with the structurally stable ZIF-8 (space group: *I-43m*, cubic), the as-derived NPCs were all featureless in microtexture (Supplementary Fig. 1). Here, upon dispersing cubic ZIF-8 in aqueous NaCl solution and subsequent evaporation, we found that the cubic ZIF-8 characterized by a rhombic dodecahedra texture transforms into the monoclinic phase with elongated close-to-hexagonal plates (Supplementary Figs. 2, 3 and Supplementary Note 2). It should be noted that the cubic and monoclinic ZIF-8 show similar chemical bonding schemes in their structure, while the crystal structure undergoes symmetry changes (Fig. 1 and Supplementary Fig. 4) during contact with NaCl solution. The sample pyrolyzed at 950 °C (termed as MCC-950) features a preserved morphology of hexagonal plates, a heritage of the monoclinic ZIF-8 crystals (Supplementary Fig. 5). Different to traditional NPCs, the powder X-ray diffraction (PXRD) pattern confirms the successful preparation of a crystalline carbon hybrid phase coming with a series of sharp peaks at 4.9°, 9.8°, 14.8°, 24.8°, 29.9°, and 35° (Fig. 2a and Supplementary

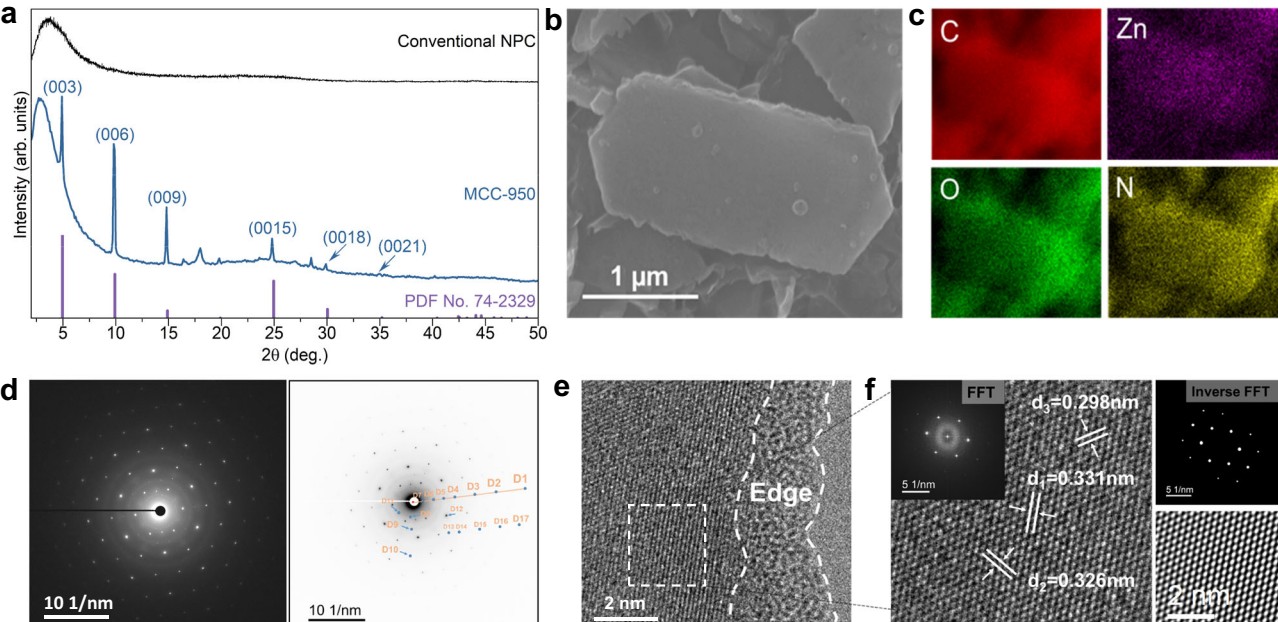

**Fig. 2 | Characterizations of MCC-950. a** XRD patterns of the final MCC-950 and conventional NPC. **b, c** SEM image (**b**) and the corresponding EDS mapping results (**c**). **d** SAED pattern (left panel) and invert contrast of electron diffraction pattern (right panel with marked spots from D1 to D11) of MCC-950. **e, f** High-resolution TEM images, and corresponding (inverse) FFT patterns (inset of **f**). Source data are provided as a Source Data file.

Table 2), which are indexed as the (003), (006), (009), (0015), (0018) and (0021) planes, similar to the structures that are found in graphite intercalation compound (graphite nitrate, PDF No.: 74-2329), respectively. This points to a very regular lamellar stacking behavior in MCC-950, with the c-direction being characterized by a large interlayer spacing(space group: *R-3m*, Rhombohedral)[36]. Note that most graphite intercalation compounds are not stable at ambient conditions due to the desorption of intercalants. In contrast, MCC-950 can well retain its intercalated structure upon exposure in air atmosphere for at least one year, as reflected by the well-preserved XRD pattern (Supplementary Fig. 6*)*. This is due to the fact that the intercalant of MCC-950 is presumably a $ZnO_x$ or $ZnN_x$ species, as discussed in detail below. The other peaks centered within 15–20° and 25–30° in the XRD pattern (Supplementary Fig. 7) are indicative of the fine structures inside the stacking layers, which cannot be directly assigned to common Zn compounds or known carbon materials.

Scanning electron microscope (SEM) images and corresponding energy dispersive spectroscopy (EDS) results (Fig. 2b, c) suggest a homogeneous distribution of the constituent elements (C, N, O, Zn) at the micrometer scale. The crystalline nature of MCC-950 is also unambiguously identified by the transmission electron microscope (TEM) images. Specifically, the selected-area electron diffraction (SAED) pattern reveals a typical hexagonal spot pattern (Fig. 2d and Supplementary Fig. 8), in which the distances of D7, D8 and D11 were measured to be 0.313-0.552 nm, matching well with the in-plane peaks of the XRD patterns (Supplementary Fig. 7 and Supplementary Table 2). This also indicates that the complex hexagons patterns are likely to be the result of the reflections from the chemically ordered atomic arrangements within the intercalation layers, and Zn is indeed the strongest scatterer in the system. Along with the lattice fringes observed on hexagonal plates (Fig. 2e, f), the corresponding fast Fourier transform (FFT) patterns further confirm the typical hexagonal arrangement of (probably Zn) atoms. The high-resolution TEM (HR-TEM) images in Fig. 2f and Supplementary Fig. 9a both identify the separable lattice fringes with hexagonal organization in MCC-950, which correspond to the periodicities of ~0.330 nm and ~0.360 nm, rather typical for ZnO substructures[37,38]. We however note that

resolving the specific carbon structure of MCC-950, because of the much weaker contrasts, is still challenging. The thermal gravimetric analysis (TGA) shows a ~80 wt% residual at 1000 °C, implying the limited mass loss of the MCC-950 at high temperatures (Supplementary Fig. 10). Indeed, high order usually induces a higher thermal stability. In the HR-TEM images, we always observe that the central region of the particles shows clearly identifiable lattice fringes, while the particle structure of MCC-950 terminates with 'amorphous' edges (Fig. 2e). We also identified amorphous area at the edge of MCC-950, which is likely to be caused by the absence of Zn-containing species as intercalants to orderly separate the carbon layers (Supplementary Fig. 9b).

Assuming a zinc-carbon intercalation compound, we performed acid (1 M HCl) leaching experiment on the samples, which resulted in a loss of ~70 wt% Zn. Here, we notice that the most intense (00 *l*) peaks of pristine MCC-950 vanished after leaching, while a strongest peak centered at 18° is identified in the XRD pattern of MCC-950-HCl (Supplementary Fig. 11a). This corresponds to a periodicity of 0.49 nm upon Zn removal. We again observe the separable lattice fringes with *d* = 0.59, 0.24 and 0.28 nm (Supplementary Figs. 11b–e) in HR-TEM images, which can be well assigned to the weak diffraction peaks centered at 15°, 37°, and 32°, respectively. We relate all these to the periodicities within the former 2D layers which as such lie in these pictures parallel to the surface. As reflected by a thorough comparison of the etched sample MCC-950-HCl (Zn: 1.5 wt% *vs.* 5.0 wt% in MCC-950), the origin of (00n) peaks in MCC-950 is clearly due to zinc species within a strictly layered carbon structure.

The highly ordered structure along *c* direction and the large layer thickness in the nanometer region hints the existence of interacting charged species at the surface in each and in different layers. It is therefore speculated that the encouraging temperature stability of MCC-950 is largely sustained by zinc linkages, which also help establish the highly ordered 2D structure by residing in between the layers. While the presence of heteroatoms (N, O, and Zn) inevitably disrupts the high order of the basal plane (i.e. *ab* plane), it is noteworthy that a less-ordered carbon framework in the *ab* direction does not necessarily imply stacking disorder in the *c* direction.

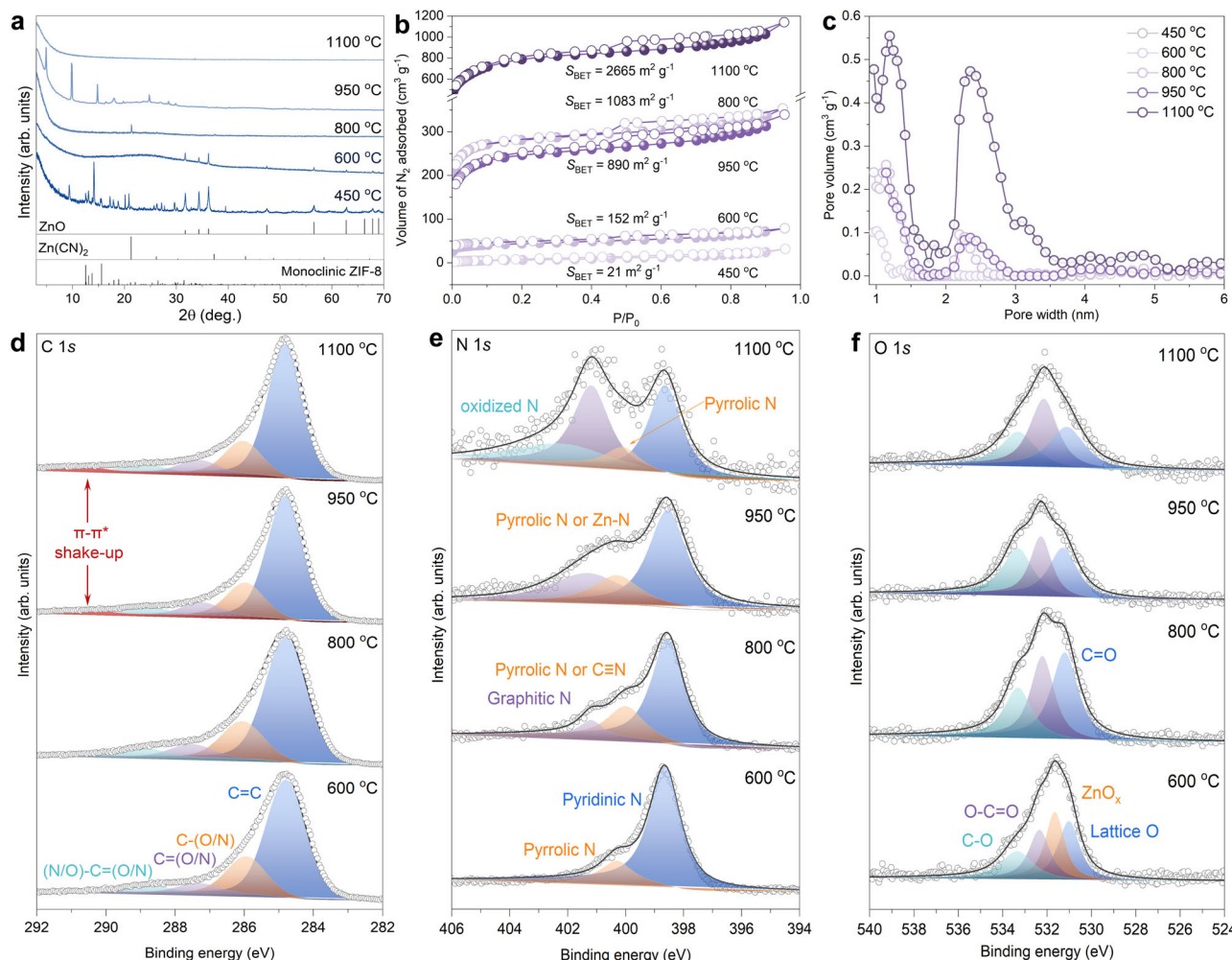

**Fig. 3 | Formation process of the samples pyrolyzed at different temperatures (T).** T = 450 °C, 600 °C, 800 °C, 950 °C, 1100 °C. **a** XRD patterns. **b** Nitrogen sorption isotherms, $S_{BET}$ represents the specific surface area. **c** Pore size distributions. **d–f** The C 1s (**d**) N 1s (**e**) and O 1s (**f**) XPS spectra with deconvoluted peaks. Source data are provided as a Source Data file.

## Structural evolution during the pyrolysis process

To elucidate the structural evolution during the pyrolysis process, the precursor was thermally treated at different temperatures. In detail, the monoclinic ZIF-8 partially transforms into zinc oxide (ZnO) at 450 °C, accompanied by the morphological change which turns from hexagonal plates into shapeless aggregates (Fig. 3a and Supplementary Figs. 12a, b). Electron Paramagnetic Resonance (EPR) spectra (Supplementary Fig. 12c) with a g value centered at 1.967 correspond to the oxygen defects of ZnO at 450 °C. Such a phase transformation appears to be completed at 600 °C, along with the regeneration of ordered hexagonal plates. Further elevating the temperature to 800 °C leads to the formation of zinc cyanide (Zn (CN)$_2$, Supplementary Fig. 13), with the preservation of plate-like microtexture. This transformation implies that zinc is more likely to coordinate with N and/or O, residing within the interlayers of the carbon structure to immobilize the MCC-950 framework. The combined effects can explain the relatively high crystallinity and long-range order of MCC-950, even it comes with high doping levels of heteroatoms (Supplementary Table 3). Note that all the diffraction peaks of MCC-950 disappear at a higher temperature of 1100 °C, indicating the collapse of the long-range ordered nanostructure at this final point. At this stage, we see additionally a thermal exfoliation effect, as the microstructure is characterized by thin-layer components. The substantial loss of Zn (decreased to 0.27 wt%) is assumed to be the primary reason for the structural collapse at 1100 °C (Fig. 3a and Supplementary Table 3). As clearly shown in Fig. 3b, the

specific surface area ($S_{BET}$) of these samples increases monotonously with elevated carbonization temperature, except for the case in MCC-950 (890 m$^2$ g$^{-1}$), which falls between that of 800 °C (1063 m$^2$ g$^{-1}$) and 1100 °C (2665 m$^2$ g$^{-1}$) samples, indicating potential variation of the interlayer porosity held by zinc-containing species. The high $S_{BET}$ value of the 1100 °C sample points to the formation of more defects and micropores. The pore size distribution of these samples highly resembles each other, albeit with substantially different crystal structure and $S_{BET}$ (Fig. 3c). Specifically, they feature a bimodal distribution of micropores (1.2 nm) and mesopores (2.4 nm), whose volumes scale with carbonization temperature and reach the highest value at 1100 °C. The combined findings are unambiguously indicative of the formation of a highly ordered packing structure of MCC-950 with internal pores in the layers, which is seemingly supported by a suitable dosage of Zn.

X-ray photoelectron spectroscopy (XPS) analyses confirm the presence of different nonmetal sites in MCCs, as shown in Fig. 3d–f and Supplementary Fig. 14. The deconvoluted C 1s XPS spectra of the 600 °C and 800 °C samples comprise four major peaks centered at 284.8, 286.0, 287.4, and 288.6 eV, corresponding to C = C, C–O (or C ≡ N), C = O, O–C = O (or N–C = N), respectively[39,40]. The new π–π shake-up satellite peak in higher-temperature (950 °C and 1100 °C) pyrolyzed carbons centers at ~290.6 eV (Fig. 3d)[41], pointing to the existence of conjugated or aromatic systems. The two major peaks in the deconvoluted N 1s XPS spectrum of the 600 °C sample are centered at 398.6 and 400.2 eV, corresponding to pyridinic (C = N–C) and

pyrrolic N (C−NH)[42], respectively (Fig. 3e). A new peak at around 399.9 eV is assigned to C ≡ N in 800 °C sample, which is related to the formation of Zn (CN)$_2$ and partly overlaps with that of pyrrolic N[43]. A comparison of the 800 °C and 950 °C samples hints that Zn primarily coordinates with pyrrolic N, as indicated by its binding energy shift of ~0.3 eV and based on the fact that Zn (CN)$_2$ is decomposed at 950 °C[44]. Zn removal at 1100 °C leaves the pyrrolic N more easily identifiable. It is also evident that higher pyrolysis temperature contributed to a higher proportion of graphitic N at 401.2 eV, along with the formation of oxidized N at 1100 °C. For the oxygen species, the O 1s XPS spectra of the 800 °C, 950 °C and 1100 °C samples give three similar signals centered at around 531.2 eV, 532.2 eV, and 533.3 eV, corresponding to C = O, O−C = O and C−O functional groups[45], respectively (Fig. 3f). The peaks centered at around 531.0 eV and 531.7 eV in lower-temperature-pyrolyzed (600 °C) sample are related to the oxygen species (Zn-O$_x$ and lattice O) in ZnO, which are consistent with XRD and EPR results.

The Zn K-edge X-ray absorption near-edge structure (XANES) spectra in Supplementary Fig. 15a indicate that the absorption edge position of MCC-950 situated in close proximity to that of ZnO and zinc phthalocyanine (ZnPc), the valence state of Zn in MCC-950 was accordingly estimated to be +2, keeping in line with the XPS results. The Fourier transform (FT) extended X-ray absorption fine structure (EXAFS) spectrum for MCC-950 (Supplementary Fig. 15b) signifies the atomic dispersion of Zn atoms due to the absence of Zn-Zn bond (2.28 Å)[46]. Instead, a distinct peak assigned to Zn-N or Zn-O (1.5 Å)[46] scattering path was identified, compelling evidence to validate the coordination of Zn with N or O in the sample (Supplementary Fig. 15c and Supplementary Table 4). The chemical environments of non-metallic elements (C, N, and O) were further investigated by the near-edge X-ray absorption fine structure (NEXAFS) spectroscopy. As shown in the C K-edge NEXAFS spectrum of MCC-950 (Supplementary Fig. 16a), the peaks at 285.5, 286.8, and 287.5 eV are assigned to π* (C = C), π* (C−OH or C−O−C) and π* (N−C = N or C = O) resonances, respectively[47]; while the broad peak centered within 290−295 eV originates from σ* (C-N) resonance[47,48]. This indicates the dominance of $sp^2$-hybridized conjugated carbon-based framework coupled with the formation of functional groups enabled by heteroatom (C and N) doping. In the N K-edge NEXFAS spectrum, the peaks at 399.8, 401.3 and 407.7 eV may originate from π* (pyridinic N), π* (pyrrolic N) and σ* (C-N) resonances[48,49], respectively. The characteristic peak assigned to metal-pyridinic N bonding generally locates between that of pyridinic N and pyrrolic N[49,50], which is however not identified in our case (Supplementary Fig. 16b). The observed asymmetrical peak at 401.3 eV may overlap with that of Zn-pyrrolic (not deconvoluted here), which is also supported by the binding energy shift of pyrrolic N in the XPS N 1s spectrum of MCC-950 when compared with other samples (Fig. 3e). The O K-edge spectrum shows a set of peaks at 530-535, 540.4 and 542-545 eV, which are indicative of π* (C−O−C or O−C = O), σ* (C-O) and σ* (C = O) resonances, respectively (Supplementary Fig. 16c)[47,51]. These results keep good consistency with XPS analyses, which adds further information of the local coordination structure of the involved elements in MCC-950, again corroborating that the highly crystalline $sp^2$-hybridized carbon network is characterized by high doping levels of heteroatoms N and O. In 600 °C sample, the bands shown in the Raman spectrum are not readily assigned to the typical D and G bands that are otherwise found in NPCs (Supplementary Fig. 17) with high graphitization degree. The collected information implies a higher oxidized or more general electron-poor state in MCC-950 framework, which is beneficial for the interaction between electron-rich substrates and the specific surface of MCC-950 during the later catalytic process.

## Catalytic performance and identification of reactive oxygen species in Fenton-like reaction

Alleviating the escalating complexity of water contaminants by new techniques is pivotal to afford accessible water resources. In this regard, Fenton-like reaction enabled by highly efficient catalysts lies at the frontier to degrade refractory pollutants, which are less likely to be removed by conventional biological treatment methods. Unlike the first- and second-generation amorphous NPCs, the MCC-950 with long-range ordered structure may come with improved electronic properties to kinetically favor chemical reactions. On the other hand, the high doping level of heteroatoms (14.2 wt% N and 9.6 wt% O in our case), which survived the high carbonization temperature, find their use in the construction of active sites within the carbon framework. Bearing these facts in mind (Supplementary Note 3), we use MCC-950 as a heterogeneous catalyst in Fenton-like reaction by the advanced oxidation process. The Fenton-like catalytic activity of MCC-950 was systematically evaluated in the degradation experiments of a broader range of pollutants (Fig. 4a, and Supplementary Table 5) upon peroxymonosulfate (PMS) activation. The dyes (methylene blue, Rhodamine B and acid orange 7) as typical chromatic pollutants from three different chemical classes, but could all be completely removed within around 5 s. Also, a 100% removal efficiency of the four phenolic compounds (phenol, bisphenol A, p-nitrophenol and 4-hydroxybenzoic acid) and two inert antibiotics (ciprofloxacin and tetracycline) were achieved within around 90 s. This comparably faster kinetics in degradation experiments (Supplementary Fig. 18) entitled MCC-950 to be one of the best catalysts for pollutant removal from water (Fig. 4b and Supplementary Table 6). Given the presence of zinc in MCC-950, we compared this work with existing single-atom zinc catalysts and comprehensively excluded its role as a potential active site in terms of catalytic activity and mechanism (Supplementary Note 4 and Supplementary Figs. 19, 20). Traditional NPCs measured as a reference show a significantly poorer ability to adsorb and degrade the pollutant (Supplementary Fig. 21). Following a 2-min degradation period, we evaluated the total organic carbon removal rate by analyzing the liquid sample separated from catalyst powder. The results, as depicted in Supplementary Fig. 22, illustrate that our system demonstrates exceptional mineralization capability, with 44-64% of organic carbon being effectively removed in just 2 min of treatment.

We further found that the zeta potentials of all samples in H$_2$O (Fig. 4c) were positive, while MCC-950-HCl possessed rather negative zeta potential after acid leaching, suggesting the removal of Zn cations from the surface and between the layers leaves the carbon framework negatively charged. The macro-cationic yet electron-poor MCC-950 allows extra interaction with foreign substrates to afford high catalytic performance.

The control experiments of the sole addition of PMS alone, MCC-950 alone, and Zn$^{2+}$ alone do not substantially decompose pollutant (Supplementary Fig. 23a). It is noteworthy that the higher surface area of the sample carbonized at 1100 °C does not contribute to a superior catalytic performance, as evidenced by its similar degradation profile with that of MCC-950 (Supplementary Fig. 23b). This is emphasized as it implies the relevant catalytic performance is bound to the crystalline nanostructure in MCC-950, relative to the already disordered nature of the sample at 1100 °C. The practically relevant catalytic performance of MCC-950 is further assessed by probing the RhB, CIP, and TC degradation kinetics in manually configured solutions (Supplementary Figs. 23c−f and Supplementary Figs. 24, 25) with different pH and interfering substances. Specifically, anions (Cl$^-$, NO$_3^-$ and HPO$_4^-$), cations (Na$^+$, Zn$^{2+}$, Mg$^{2+}$, and Ca$^{2+}$), hardness, and the presence of humic acid (as a soil model) did not significantly influence the degradation rate of these pollutants. These experimental outputs show that the catalytic degradation initiated by MCC-950 is universally applicable and less likely to be intercepted by common organic and inorganic substances in water. We additionally monitored the corresponding zinc ion leaching after using MCC-950 for activation reaction under different initial pH conditions. The results, as illustrated in Supplementary Figs. 26, demonstrate that MCC exhibits minimal zinc ion leakage across a broad pH spectrum, ranging from pH 1 to 9.

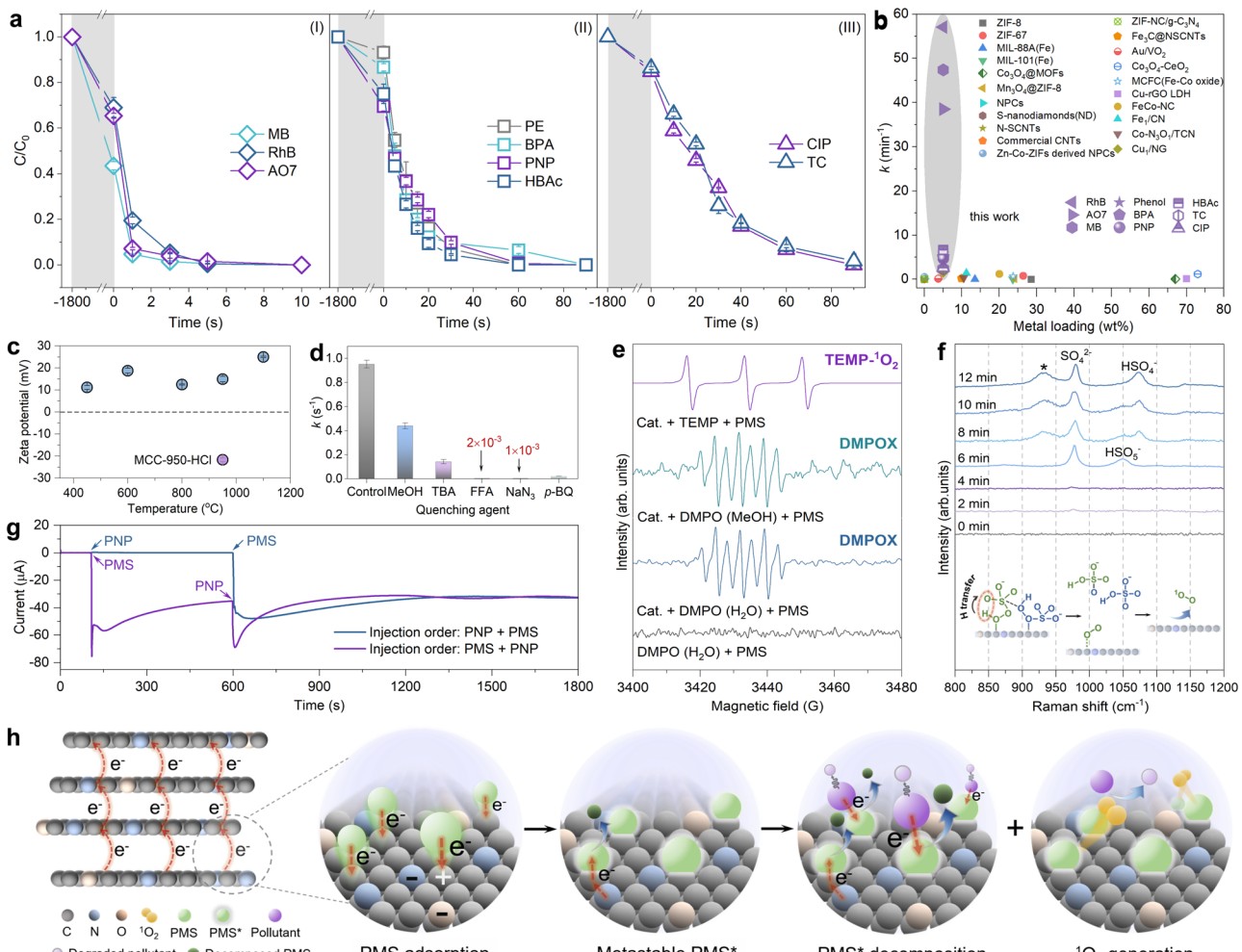

**Fig. 4 | Fenton-like catalytic performances of MCC-950. a** Degradation profiles of MCC-950 towards dyes (I), phenol derivatives (II) and antibiotics (III) upon PMS activation. MB: methylene blue, RhB: Rhodamine B, AO7: acid orange 7, PE: phenol, BPA: bisphenol A, PNP: p-nitrophenol, HBAc: 4-hydroxybenzoic acid. **b** A comparison of the catalytic degradation performance with recently reported cases. **c** Zeta potentials of MCC-950 and the acid-etched sample MCC-950-HCl. The shadowed regions (light gray color) in (**a**–**c**) denote the adsorption period (30 min) prior to catalytic reaction. **d** Comparison of quenching kinetics of MCC-950 under different conditions. **e** EPR spectra for PMS activation using different probing agents. **f** In situ Raman spectra of MCC-950/PMS system. Inset: the binding and conversion processes of PMS molecules on the catalyst surface. **g** Current response of MCC-950 upon changing the injection sequences of PMS and PNP. **h** Proposed PMS activation mechanism at the interfaces of MCC-950, which is adsorbed and then transforms into metastable intermediate PMS*, followed by decomposition and the selective generation of $^1O_2$. Reaction condition: [pollutants] = 20 mg L$^{-1}$, [PMS] = 0.4 g L$^{-1}$, [catalyst] = 0.08 g L$^{-1}$, $T$ = 298 K, initial solution pH = 6.5. Error bars in (**a, c, d**) represent the standard deviations of three independent measurements. Source data are provided as a Source Data file.

Importantly, these levels remain well below the WHO-established safe reference value of 3 ppm[52]. The cycling tests further verify the encouraging stability of MCC-950 (Supplementary Fig. 27).

To identify the possible reaction pathways in MCC-950/PMS system, quenching experiments were performed (Fig. 4d). Clearly, methanol and t-butyl alcohol exerted limited impacts on RhB degradation, while furfuryl alcohol and NaN$_3$ completely quenched the reaction, suggesting that singlet oxygen ($^1O_2$) is a relevant reactive oxygen species (ROS) upon PMS activation. EPR experiments were further conducted to identify the ROS in MCC-950/PMS system (Fig. 4e). By using DMPO (5,5-dimethyl-1-pyrroline N-oxide, dissolved in H$_2$O) as probing agent, we observed typical septet signals that are assigned to DMPOX (5,5-dimethyl-1-pyrrolidone-n-oxyl), which serves as a product of the non-radical pathways in PMS activation (Supplementary Fig. 28)[53–55]. This is further backed by using MeOH as solvent, which can capture superoxide radicals (O$_2$$^{•−}$) signals if they are produced. However, we found that the DMPOX signals remain unchanged, underlining the non-radical pathways to be dominant in MeOH solvent. When TEMP (2,2,6,6-

tetramethylpiperidine) was used, the triplet peaks of TEMP-$^1O_2$ were clearly detected in the EPR spectrum. We further conducted solvent exchange experiments, replacing the solvent H$_2$O with D$_2$O (Supplementary Fig. 29). When D$_2$O was used as the solvent, we observed a clear enhancement in the degradation kinetics of the MCC-950/ PMS/Pollutant system, thereby confirming $^1O_2$ as dominant ROS in the system. Next, in situ Raman spectra were recorded to reveal the surface chemical evolution in MCC-950/PMS (Fig. 4f). A dominant peak centered at ~980 cm$^{-1}$, which is assigned to SO$_4$$^{2-}$, gradually emerged with prolonged reaction time. The species located at ~930 cm$^{-1}$ is primarily ascribed to the formation of peroxo species bonded to the surface sites, and their stretching vibrations. The intermediate species (HSO$_5$$^−$) centered at ~1050 cm$^{-1}$ gradually disappears[56], along with the appearance of a new peak at 1075 cm$^{-1}$, which implies the transformation from HSO$_5$$^−$ to HSO$_4$$^−$[57]. The transformation from HSO$_5$$^−$ to SO$_4$$^{2-}$ and HSO$_4$$^−$ indicates the hydrogen transfer process which finally generates $^1O_2$. The above results collectively suggest that MCC-950 can efficiently activate PMS for improved advanced oxidation processes.

## Catalytic mechanism

Chronoamperometry measurements were performed to monitor the surface-activated intermediates and electron flow during PMS activation. The injection of PMS gave rise to distinct current drop and the largest current gap was detected in MCC-950 (Fig. 4g and Supplementary Figs. 30a, b), which signifies the promoted electron transfer in the highly crystalline carbon. Clearly, there were two stages of current changes after adding PMS. The first stage came with rather steep tendency and points to the fact that the adsorbed PMS can rapidly assign electrons to the electron-poor MCC-950 with long-range ordered structure, accompanied by the transformation from PMS to its metastable intermediate PMS* with accepting electrons (current drop). The activation reaction on the catalyst surface becomes sluggish due to the significant occupation of active sites and constraints in mass transfer within the solution (current rebound). Benefiting from the high doping amounts of the heteroatoms N and O with lone-pair electrons, PMS* is readily activated by Lewis-base sites to decompose quickly. Following the consumption of PMS* at the first stage, the adsorption sites in MCC-950 are regenerated to further capture PMS, albeit slower in the second stage, indicating that the catalyst is still loaded with electrons. Among the four phenol derivatives as pollutants, PNP shows the poorest electron-donating ability. Therefore, we used PNP as a comparably inert probe to evaluate the efficiency of deep oxidation. The injection of PNP again resulted in a current drop and indicates the electron transfer from PNP to PMS*, while the decomposition of PMS* explains the current rebound. During this process, the activated pollutant was subjected to an advanced oxidation process for catalytic degradation. Note that when PNP was added at first, no current response was observed, which implies the absence of electron transfer from PNP to catalyst surface and highlights the importance of PMS activation for pollutant degradation. This process was further evidenced by a noticeable change of current in the linear sweep voltammetry curve of MCC-950/PMS (Supplementary Fig. 30c), which reflects the electron transfer from the catalyst to the adsorbed PMS. We further evaluated the consumption of PMS with or without the pollutant, and the results showed that the introduction of pollutant increased the utilization rate of PMS (Supplementary Fig. 31) from 46% to 60%. This can be directly related to the effective electron transfer to the targets which then consumes the surface-adsorbed PMS.

It was previously discussed that the stoichiometry between surface-activated PMS and pollutants could be related to the origin of catalytic activity for N- and O-doped carbon materials[58,59]. In general, lower carbonization temperature contributed to more rapid stoichiometry-dependent reaction at a constant electron transfer rate, while higher temperatures gave fewer binding sites, which is fortunately compensated by higher kinetics and catalytic performance. As discussed above, the well-preserved atomic arrangement of MCC-950 enables not only a very high density of active sites, but also the necessary high-speed electron transfer. The electron-deficient, heteroatom-doped carbon effectively promotes the primary activation of negative PMS, which then enables a nucleophilic attack onto the elongated S-O bond to form an activated PMS* with dramatically improved oxidative capability. As intermediate species, surface-confined PMS* might oxidize the pollutants at the interface of MCC, or it generates interfacial $^1O_2$ (Fig. 4h). Combined with the carbon network around active centers and the nanoscopic 2D layered structure of MCC-950, the electron transfer from pollutants is not limited to the close proximity of the surface-adsorbed PMS itself, but the total carbon surface.

In summary, we conclude that interfacial $^1O_2$ and surface-activated PMS are the main oxidative species for the degradation of pollutants that occur on the interfaces of the catalyst. The electron-deficient MCC modulated by N- and O-doping in carbon frameworks promote the primary adsorption of PMS, while electron transfer as the second contribution to binding gives surface-activated oxidative PMS species, and this activation process is highly promoted by the rapid transfer of electrons and efficient electron delocalization due to the highly crystalline nanostructure around active centers (Fig. 4h). All these well explain the fast dynamics of MCC-950 compared with other types of catalysts.

## Discussion

To summarize this work, we used monoclinic-ZIF-8 as precursor and NaCl as closed salt template to synthesize a MOF-derived crystalline carbon (MCC). Contrary to the first- and second-generation NPCs, our system is characterized by high internal long-range order, and one might describe it as an N- and O-containing carbonaceous crystal obtained from organics through thermal condensation, to our opinion the first case of a third-generation MOF-derived NPCs. Although the local atomistic structure could not be revealed in all details, the material is found to be unusually stable at temperatures up to 1000 °C and chemically highly functional and active. The combined characterizations verified the highly ordered 2D structure of MCC as a layered carbon-Zn hybrid material, very similar to the structure of some graphite intercalation or intermetallic compounds. It is characterized by perfectly two-dimensional, negatively charged porous carbon layers with oxy- and imide surfaces, Coulomb-stabilized by residing Zn-containing species in between the layers. The structure is potentially best described as a 2D-zeolitic system made up of C, N, O, and Zn. Our initial attempts to apply MCC in Fenton-like advanced oxidation process witnessed a high catalytic activity of MCC, with a complete removal of a broad range of reference pollutants, which are decontaminated within a contact time at the seconds scale. Beyond environmental remediation, MCC is also expected to be a promising candidate for other carbocatalytic applications, electrocatalytic energy storage and conversion, electronics, and beyond.

## Methods

### Chemicals and reagents

Zinc nitrate hexahydrate (98%, reagent grate), 2-methylimidazol (≥ 98%), sodium chiloride (≥ 99.5%) ciprofloxacin (≥ 98%, CIP), and Rhodamine B (RhB, 98%) were purchased from Fisher Scientific. Potassium hydrogen monopersulfate (Oxone, $2KHSO_5 \cdot KHSO_4 \cdot K_2SO_4$), sodium hydroxide, sodium thiosulfate, sulfuric acid, potassium iodide (≥ 99%), dimethyl sulfoxide (DMSO), furfuryl alcohol (99.8%, FFA), benzoquinone (99%, BQ), tert-butyl alcohol (≥99.5%, TBA), p-nitrophenol (99%, PNP), methyl blue (99%, MB), acid orange 7 (≥85%, AO7), phenol (≥99%, PE), bisphenol A (97%, BPA), 4-hydroxybenzoic acid (99%, HBA), absolute ethanol (99.9%), potassium iodide (99%, KI), humic acid (98%, ~20% ign. residue), Deuterium oxide (99.9% D, $D_2O$) and 2,2,6,6-tetramethylpiperidine (≥ 99%, TEMP) were purchased from Merck Schuchardt OHG. The 5,5-dimethyl-1-pyrroline N-oxide (≥98%, DMPO) and tetracycline (≥95%, TC) were obtained from TCI Co., Ltd. All chemicals were of analytical grade and used without further purification.

### Salt-assisted preparation of monoclinic-ZIF-8

The cubic ZIF-8 powder was synthesized in MeOH at room temperature. Specifically, 2.14 g zinc nitrate hexahydrate and 4.71 g 2-methylimidazol were individually dissolved in 80 mL MeOH, which were then quickly mixed and stirred for 1 h, followed by aging for 24 h. The obtained white crystals were collected by centrifugation and washed with MeOH several times, and finally dried at 60 °C overnight under dynamic vacuum. Subsequently, the cubic ZIF-8 power was dispersed in aqueous NaCl solutions (3.2 g in 10 mL). After ultrasonication for 1 h, the suspension was further stirred for 24 h at 1200 rpm. Then, the aqueous suspension was evaporated at 80 °C in oil bath with stirring for 24 h to yield white powder.

### Salt-assisted carbonization of MOF precursors and NPCs

The as-prepared monoclinic ZIF-8 was directly carbonized in $N_2$ atmosphere at different temperatures. Specifically, the precursor was

firstly heated to 350 °C and kept for 1 h, then further heated to target carbonization temperature ($T = 400, 600, 800, 950,$ and $1100$ °C) with a ramping rate of 3 °C/min and hold for 2 h. Upon cooling down to room temperature, the powder was washed with deionized water and dried at 60 °C overnight to obtain MCC-950 or other products at different temperatures ($T = 400, 600, 800,$ and $1100$ °C). NPC was prepared by the same heating program using cubic ZIF-8 as precursor. To reveal the role of salt in the hydrolysis process, another NPC was synthesized using a mixture of cubic ZIF-8 (100 mg) and NaCl (3.2 g), following the same pyrolysis procedures as described above.

## Characterizations

SEM images were recorded on a LEO 1550-Gemini instrument after sputtering with platinum. XRD patterns were recorded with a multi-purpose X-ray Diffractometer (Rigaku, SmartLab) equipped with a 3 kW X-ray generator and a long-focus 2.2 kW Cu X-ray tube. The step size used for measurements was 0.1 ° min$^{-1}$. The TEM study was performed using a double Cs corrected JEOL JEM-ARM200F (S)TEM operated at 80 kV and equipped with a cold-field emission gun and a high-angle silicon drift Energy Dispersive X-ray (EDX) detector. The Raman spectra of solid catalysts were obtained using a Confocal Raman Microscope (Alpha 300 R, WITec) and the wavelength of laser was 785 nm. The in situ Raman spectra were performed using a LabRAMHR Evolution Raman microscope with laser excitation at 514.5 nm at room temperature. Nitrogen sorption isotherms were recorded at 77 K after activating the samples at 150 °C under vacuum for 24 h with a Quantachrome Quadrasorb SI porosimeter. The specific surface area was determined by Brunauer–Emmett–Teller (BET) method, while the pore size distribution analysis was completed with QSDFT model using the adsorption branch ($P/P_0$ range: $1.00e^{-7}$ to $1.00$) and QuadraWin software. Surface elemental compositions of materials were investigated via XPS system equipped with a monochromatic Al Ka line (1486.7 eV). A X-Band EPR spectrometer was utilized for the radical analyses. The FT-IR spectra were measured with a Thermo Scientific Nicolet iS5 FT-IR spectrometer from 4000 to 400 cm$^{-1}$. UV–Vis spectra were measured using a UV-2600 spectrophotometer (Shimadzu, Japan). The measurements of total organic carbon (TOC) were conducted after the separation of catalyst at 2 min using TOC-L series analyzer (Shimadzu, Japan). The electrochemical tests were carried out on a Gamry workstation using the physical chemistry module. The three-electrode cell comprised a Pt wire as the counter electrode, an Ag/AgCl electrode as a reference electrode and a glassy carbon modified with the catalyst (diameter: 3 mm) as the working electrode. To prepare the ink, 1 mg catalyst was mixed with 100 μL deionized water, 100 μL EtOH and 40 μL Nafion 117 solution (5 wt%, Aldrich). Subsequently, 10 μL of catalyst ink was dropped on the glassy carbon tip and dried at room temperature. The chronoamperometry and LSV tests were performed in 50 mL 0.1 M Na$_2$SO$_4$ solution as supporting electrolytes.

C, N, and O K-edge NEXAFS spectra were recorded at the photoemission end-station (BL10B) beamline of the National Synchrotron Radiation Laboratory (NSRL) in Hefei, China. Zn K-edge analysis was performed with Si (111) crystal monochromators at the BL11B beamlines at the Shanghai Synchrotron Radiation Facility (SSRF) (Shanghai, China). Before the analysis at the beamline, samples were pressed into thin sheets with 1 cm in diameter and sealed using Kapton tape film. The XAFS spectra were recorded at room temperature using a 4-channel Silicon Drift Detector (SDD) Bruker 5040. Zn K-edge extended X-ray absorption fine structure (EXAFS) spectra were recorded in transmission mode. Negligible changes in the line-shape and peak position of K-edge XANES spectra were observed between two scans taken for a specific sample. The XAFS spectra of these standard samples (ZnPc, ZnO, and Zn foil) were also recorded in transmission mode. The acquired EXAFS data were extracted and processed according to the standard procedures using the ATHENA module implemented in the IFEFFIT software packages[60].

Subsequently, $k^3$-weighted $\chi(k)$ data in the k-space were Fourier transformed to real R space using a hanning windows (d$k = 1.0$ Å$^{-1}$) to separate the EXAFS contributions from different coordination shells. To obtain the quantitative structural parameters around Zn atoms in the sample, least-squares curve parameter fitting was performed using the ARTEMIS module of IFEFFIT software packages[61]. Effective back-scattering amplitudes $F(k)$ and phase shifts $\Phi(k)$ of all fitting paths were calculated by the ab initio code FEFF8.0[62]. For the sample, a $k$ range of 2.9–12.45 Å$^{-1}$ was used and curve fittings were done in the R-space within the R range of 1–2 Å for $k^3$-weighted $\chi(k)$ functions. The number of independent points is

$$N_{ipt} = \frac{2 \times \Delta k \times \Delta R}{\pi} = \frac{2 \times (12.45 - 2.9) \times (2 - 1)}{\pi} \approx 6.08$$

For this sample, the Fourier-transformed curves showed two single prominent coordination peaks at 1.5 Å assigned to the Zn-N/O coordination. During the curve fitting for the sample, Debye–Waller factors ($\sigma^2$), coordination numbers (CN), interatomic distances ($R$) and energy shift ($\Delta E_0$) were treated as adjustable parameters for the Zn-N path. The number of adjustable parameters was $N_{para} = 1 + 1 + 1 + 1 = 4$, less than the $N_{ipt}$.

## Batch experiments

The degradation reactions were performed in a 100 mL flask at room temperature ($25 \pm 3$ °C) with magnetic stirring at 500 rpm. Unless stated otherwise, the initial concentrations of pollutants (MB, RhB, AO7, PE, BPA, PNP and HBAc), PMS, and catalysts were 20 mg L$^{-1}$, 0.4 g L$^{-1}$, and 0.08 g L$^{-1}$, respectively. Prior to the degradation process, the catalyst was dispersed in H$_2$O and sonicated for 10 min. Then, the pollutant was added to the solution and stirred for 30 min to achieve adsorption equilibrium. Subsequently, the reactions were initiated by adding PMS to the solution already mixed with the catalyst and pollutant. The reaction mixture was periodically collected and immediately quenched with excess Na$_2$S$_2$O$_3$ solution (200 mM). The mixture was then filtered through a PTFE membrane (pore size: 0.45 μm) to separate the solid catalyst, which was retained for further analysis. The initial pH was adjusted to the desired values using either 1 M H$_2$SO$_4$ or 1 M NaOH. All experiments were conducted at least three times to afford reliable data. The effects of potential influential factors, including pH, inorganic ions, hardness[63] and humic acid, were investigated by changing the parameters accordingly. The as-used catalysts were collected by filtration, washed by H$_2$O three times and died at 60 °C overnight for next catalytic cycle. In the solvent exchange experiment, D$_2$O was employed instead of deionized water, while all other procedures remained consistent with the degradation experiments described above.

The concentrations of PE and HBAc were determined by HPLC (Thermo, USA) equipped with a C18 column. A mixture of 1% acetic acid aqueous solution and MeOH (v/v = 50%: 50%) was adopted as the mobile phase with a flow rate of 0.2 mL min$^{-1}$. The detection wavelength was 270 nm for PE and 250 nm for HBAc. The concentrations of all other pollutants were determined by ultraviolet-visible absorption spectroscopy. The concentration of HSO$_5^-$ anion (PMS) was determined by a KI spectrophotometric method using UV−Vis spectroscopy. The KI solution was first prepared by dissolving 10 g KI and 0.5 g NaHCO$_3$ in 100 mL H$_2$O. The sample was filtered to collect the filtrate (125 μL), followed by adding 5 ml of KI solution to form a yellow mixture. Then, it was kept for 10 min to achieve stability of the mixed solution. Finally, the absorbance was measured at a fixed wavelength of 395 nm.

## Data availability

The experimental data generated in this study are provided in the Supplementary Information and Source Data file. Source data are provided with this paper.

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

## Acknowledgements

This work was financially supported by the Max Planck Society. Y.W. thanks the Alexander von Humboldt Foundation for a postdoctoral fellowship. The authors thank BL10B in the National Synchrotron Radiation Laboratory (NSRL, Hefei, China) for the XAFS tests.

## Author contributions

Y.W. and M.A. supervised the project. T.L. and Y.W. conceived the project and idea. T.L. conducted most of the experiments. L.X. conducted synchrotron radiation experiments and helped with data acquisition and analysis. D.P. and N.V.T. contributed to electron microscopy experiments and data processing. J.Y. helped with the X-ray photoelectron spectroscopy test and analysis. T.L. and Y.W. analyzed the data and drafted the manuscript. T.L., Y.W., and M.A. wrote the final manuscript with input from all authors. All authors discussed the results and commented on the manuscript.

## Funding

## Competing interests

The authors declare no competing interests.
