## [Peer Review File · Nature Communications]

Metal-organic framework derived crystalline nanocarbon for Fenton-like reactionEditorial note: This manuscript has been previously reviewed at another journal that is not operating a transparent peer review scheme. This document only contains reviewer comments and rebuttal letters for versions considered at *Nature Communications*.

REVIEWER COMMENTS

Reviewer #1 (Remarks to the Author):

The reviewer went through the response and changes that the authors have made, and found most of the changes remarkably precise and "on the point". The reviewer's concerns are sufficiently addressed. Thus, now I can support publication in the present form.

Reviewer #2 (Remarks to the Author):

The manuscript by Lian et al. describes the synthesis, characterization and catalytic application of a crystalline nanocarbon catalyst. A salt-assisted carbonization strategy is adopted to obtain the crystalline nanocarbon using ZIF-8 as the precursors. The authors claim that the as-prepared catalyst comes with a high level of nitrogen and oxygen terminating the 2D layers and thus shows impressive performance in Fenton-like reaction. Although the authors have made great efforts to address the concerns of the previous reviewers, I still believe that the manuscript is not suitable for publication because the accurate structure of the prepared material has not been confirmed.

1. The authors are suggested to conduct XAFS to characterize the coordination structures of the metallic and non-metallic elements in crystalline nanocarbon.
2. Theoretical calculations are suggested to simulate the proposed atomic structure of crystalline material.
3. The role of sodium chloride in the carbonization process should be considered. Can crystalline nanocarbon be obtained by directly calcining the m-ZIF-8 in the absent of NaCl? Will NaCl be remained in the obtained material? Is the residual NaCl a part of the composition of the crystalline nanocarbon?
4. Line 125, page 5. "remains stable in air atmosphere for at least one year". Is there any experimental data?

5. Line 153, page 6. It is unreasonable to use thermogravimetry to investigate the thermal stability of materials as phase transformation may also occur at high temperatures.
6. The roles of Zn and residual NaCl in the Fenton reaction should not be neglected.

Point-by-point responses to reviewers' comments

REFEREE REPORTS

Reviewer #1 (Remarks to the Author):

The reviewer went through the response and changes that the authors have made, and found most of the changes remarkably precise and "on the point". The reviewer's concerns are sufficiently addressed. Thus, now I can support publication in the present form.

Response: The authors appreciate Reviewer #1 for her/his positive feedback on our revision.

Reviewer #2 (Remarks to the Author)

The manuscript by Lian et al. describes the synthesis, characterization and catalytic application of a crystalline nanocarbon catalyst. A salt-assisted carbonization strategy is adopted to obtain the crystalline nanocarbon using ZIF-8 as the precursors. The authors claim that the as-prepared catalyst comes with a high level of nitrogen and oxygen terminating the 2D layers and thus shows impressive performance in Fenton-like reaction. Although the authors have made great efforts to address the concerns of the previous reviewers, I still believe that the manuscript is not suitable for publication because the accurate structure of the prepared material has not been confirmed.

Response: The authors appreciate the critical comments from Reviewer #2. Following the suggestions, we have tried our best to supplement XAFS analysis and other tests as well as clarifications, which can hopefully address all the concerns and merit publication in Nature Communications. The detailed responses are as follows.

1. The authors are suggested to conduct XAFS to characterize the coordination structures of the metallic and non-metallic elements in crystalline nanocarbon.

Response: We split the characterization and discussion into two parts: metallic element and non-metallic elements.

(1) Metallic element (Zn)

The Zn K-edge X-ray absorption near-edge structure (XANES) spectra in Fig. R1a indicate that the absorption edge position of MCC-950 situated in close proximity to that of ZnO and zinc phthalocyanine (ZnPc), the valence state of Zn in MCC-950 was accordingly estimated to be +2, keeping in line with the XPS results. The Fourier transform (FT) extended X-ray absorption fine structure (EXAFS) spectrum for MCC-950 (Fig. R1b) signifies the atomic dispersion of Zn atoms due to the absence of Zn-Zn bond (2.28 \AA)^[R1]. Instead, a distinct peak assigned to Zn-N or Zn-O (1.5 \AA)^[R1] scattering path was identified, a compelling evidence to validate the coordination of Zn with N or O in the sample. The N and O coordination species cannot be distinguished by EXAFS, because the peaks that correspond to the Zn-N and Zn-O bonds lie very close to each other (Fig. 1b). Taking Zn-N interaction as a typical example, the fitted results of FT-EXAFS spectrum in R space points to the Zn-N₄ configuration as the local structure of Zn in MCC-950, that is, one Zn atom coordinates with four N atoms (Fig. R1c and Table R1).

Generally, the central metal sources in the reported single-atom catalysts for catalytic oxidation processes are transition metal atoms with partially occupied 3d orbitals (e.g. Fe, Co, Cu, Mn)^[R2-R5], which are conducive to electron transfer during the reactions. Comparatively, the fully occupied 3d orbitals of Zn (*i.e.* 3d¹⁰ configuration) largely tethers the electron mobility and therefore renders Zn²⁺ intrinsically inactive for the catalytic reactions. This is reflected in the recent report in Fenton-like reaction using Zn-based single-atom catalyst, which shows substantially lower activity in the catalytic degradation of various pollutants. For example, the rate constant for the degradation of acid orange 7

(AO7) was 0.136 min^{-1} (degradation time: 30 min)^[R6], while it was determined to be 38.5 min^{-1} (degradation time: 5 s) in our case (Fig. 4b in main text).

The above information has been added in main text and Supplementary Information, and the XAFS results have been provided as new Supplementary Fig. 15.

Page 9, Lines 231-239:

“The Zn K-edge X-ray absorption near-edge structure (XANES) spectra in Supplementary Fig. 15a indicate that the absorption edge position of MCC-950 situated in close proximity to that of ZnO and zinc phthalocyanine (ZnPc), the valence state of Zn in MCC-950 was accordingly estimated to be +2, keeping in line with the XPS results. The Fourier transform (FT) extended X-ray absorption fine structure (EXAFS) spectrum for MCC-950 (Supplementary Fig. 15b) signifies the atomic dispersion of Zn atoms due to the absence of Zn-Zn bond (2.28 \AA)⁴⁶. Instead, a distinct peak assigned to Zn-N or Zn-O (1.5 \AA)⁴⁶ scattering path was identified, a compelling evidence to validate the coordination of Zn with N or O in the sample (Supplementary Fig. 15c and Supplementary Table 4).”

Fig. R1 Chemical and coordination environment of Zn in MCC-950. **a**, Zn K-edge XANES spectra of MCC-950 and reference samples (Zn foil, ZnO and ZnPc). **b**, FT k^3 -weighted Zn K-edge EXAFS spectra of MCC-950 and reference samples. **c**, FT-EXAFS fitting curve of MCC-950 at Zn K-edge.

Table R1 EXAFS fitting parameters at the Zn K-edge for MCC-950

Sample	Path	CN	R (\AA)	σ^2 (10^{-3})	ΔE_0 (eV)	R-factor
MCC-950	Zn-N	4.2	2.03	0.008	0.98	0.009

CN: coordination number; R: bond distance; σ^2 : Debye-Waller factors; ΔE_0 : energy shift; R-factor: goodness of fit.

For the benefit of the broad readership, we also summarize the zinc-based catalysts for Fenton-like reaction and discussed in Supplementary Note 4, which help distinguish previous work from the current one. The relevant discussion is also listed as follows:

Page 11, Lines 288-291 (main text):

“Given the presence of zinc in MCC-950, we compared this work with existing single-atom zinc catalysts and comprehensively excluded its role as potential active site in terms of catalytic activity and mechanism (Supplementary Note 4 and Supplementary Figs. 19-20).”

Supplementary Information:

“Generally, the central metal sources in the reported single-atom catalysts for catalytic oxidation processes are transition metal atoms with partially occupied 3d orbitals (e.g. Fe, Co, Cu, Mn)^{24,34-36}, which are conducive to electron transfer during the reactions. Comparatively, the fully occupied 3d orbitals of Zn (i.e. 3d¹⁰ configuration) largely tethers the electron mobility and therefore renders Zn²⁺ intrinsically inactive for the catalytic reactions. This is reflected in the recent report in Fenton-like reaction using Zn-based single-atom catalyst, which shows substantially lower activity in the catalytic degradation of various pollutants. For example, the rate constant for the degradation of acid orange 7 (AO7) was 0.136 min⁻¹ (degradation time: 30 min)³⁷, while it was determined to be 38.5 min⁻¹ (degradation time: 5 s) in our case (Fig. 4b in main text). Apart from the differences in performance, the catalytic mechanism of the reported single-atom zinc catalysts also differs from those observed in our work. Adjusting the ratio of nitrogen (N) to carbon (C) or oxygen (O) elements coordinated to a single Zn atom can extend the distribution of Zn 4s states near the Fermi level, this could result in stronger hybridization of Zn 4s with empty O₂ 2p* states, thereby facilitating electron transfer in the subsequent protonation process of adsorbed O₂^[R7]. Several cases have attempted to

activate zinc sites based on this principle, employing a dissolved oxygen activation mechanism to enhance the electron transfer of electron-rich pollutants to PMS with the Zn-N₄ configuration. In these studies, dissolved oxygen is reduced at the zinc site to produce superoxide radicals ($\cdot\text{O}_2^-$), which can participate in the degradation of pollutants^[R6,R8,R9]. Although $\cdot\text{O}_2^-$ from dissolved oxygen activation is only a minor oxidation pathway for degradation, the intermediate formed by the Zn center binding to PMS is considered to be a more significant active species. This dissolved oxygen activation mechanism serves as corroborative evidence that the zinc site can be activated as an active site.

In fact, the "active center" ZnN₄ is more of a binding site for PMS molecules, forming a surface complex and obtaining electrons from the electron-rich area (intermediate) around the Zn site, following a typical electron transfer path. For this surface complex mechanism, the purpose of pollutant degradation depends on the rapid electron transfer process near the active center. Yu et al. directed the synthesis of single-atom zinc sites on the edge of carbon-based carriers, which accelerated the electron transfer process by strengthening the interaction between zinc sites and edge carbon structures, further improving the catalytic degradation kinetics^[R10]. Specifically, the single-atom zinc sites on the edge are modulated by the surrounding carbon structure, and the adsorption of HSO₅⁻ on the ZnN₄-edge site is relatively weak. HSO₅⁻ is more likely to be first decomposed into H and SO₅ intermediate species on the ZnN₄-edge. The carbon-based structure near the Zn sites exhibits synergistic effect on the decomposition products of PMS molecules. Then, the nearby electron-rich N site bonds the separated H atoms with dangling bonds and stabilizes SO₅. Subsequently, the intermediate species are rapidly transformed into active species, which then participate in the degradation of pollutants. This mechanism highlights the importance of the edge structure of zinc sites, ensuring that PMS molecules are quickly transformed and activated. Despite these advancements, MCC-950 in our work demonstrates a kinetic activity 15

times higher than the best Zn-N₄-edge-NC catalysts, showcasing its superior catalytic performance. However, unlike Co metal centers, the Zn centers do not cause PMS to decompose into typical free radicals. Instead, the activation and degradation process are controlled by efficient electron transfer, reinforcing the high catalytic activity of single-atom catalysts.

About the active mechanism of this work, the EPR results excluded the formation of $\cdot\text{O}_2^-$, differing from the active species in previous studies (Figure 4e in main text). To further explore the effect of dissolved oxygen (DO) on the degradation process in the MCC-950/PMS system, we conducted PMS activation experiments after removing DO from the system. The results showed no attenuation in the degradation process (Fig. R2, also added as new Supplementary Fig. 19 in SI), indicating that DO in the MCC/PMS system does not participate in degradation process and also does not produce $\cdot\text{O}_2^-$.

As we explained in the main text regarding the ultrafast kinetic mechanism of MCC, the retention of more than 20% by mass ratio of heteroatom doping in MCC-950 induces nearby carbon atoms to exhibit electron-deficient properties, which can act as PMS capture sites. Additionally, the synergistic effect of the nearby structure ensures the rapid progress of the electron transfer pathway. This provides a reasonable explanation for the superior performance of MCC-950 compared to single-atom zinc catalysts. Furthermore, combined with the observed change in the interlayer spacing of MCC-950 after acid etching and SEM images which show thin layer morphologies (Supplementary Fig. 20), it can be inferred that zinc serves as an important intercalation agent for stabilizing the MCC structure rather than being the main active site contributing to the catalytic activity. This structural stabilization by zinc, rather than direct participation in the catalytic process, highlights the distinct mechanism through which MCC-950 achieves its high performance, emphasizing the role of the carbon structure and heteroatom doping in facilitating rapid and efficient electron transfer for PMS activation.”

Figure R2 Degradation profiles of MCC-950 towards RhB, PE and CIP upon PMS activation in normal solution (with DO) and DO-removed solution.

(2) Non-metallic elements (C, N and O)

The chemical environments of non-metallic elements (C, N and O) were further investigated by the near-edge X-ray absorption fine structure (NEXAFS) spectroscopy. As shown in the C K-edge NEXAFS spectrum of MCC-950 (Fig. R3a), the peaks at 285.5, 286.8 and 287.5 eV are assigned to π^* (C=C), π^* (C-OH or C-O-C) and π^* (N-C=N or C=O) resonances, respectively^[R11]; while the broad peak centered within 290-295 eV originates from σ^* (C-N) resonance^[R11, R12]. This indicates the dominance of sp^2 -hybridized conjugated carbon-based framework coupled with the formation of functional groups enabled by heteroatom (C and N) doping. In the N K-edge NEXAFS spectrum, the peaks at 399.8, 401.3 and 407.7 eV may originate from π^* (pyridinic N), π^* (pyrrolic N) and σ^* (C-N) resonances^[R12, R13], respectively. The characteristic peak assigned to metal-pyridinic N bonding generally locates between that of pyridinic N and pyrrolic N^[R13, R14], which is however not identified in our case. The observed asymmetrical peak at 401.3 eV may overlap with that of Zn-pyrrolic (not deconvoluted here), which is also supported by the binding energy shift of pyrrolic N in the XPS N 1s spectrum of MCC-950 when compared with other samples (Fig. 3e in main text). The O K-edge spectrum shows a set of peaks at 530-535, 540.4 and 542-545 eV, which are indicative of π^* (C-O-C or

O-C=O), σ^* (C-O) and σ^* (C=O) resonances, respectively^[R11, R15]. The combined results keep good consistency with XPS analyses, which adds further information of the local coordination structure of the involved elements in MCC-950, again corroborating that the highly crystalline sp^2 -hybridized carbon network is characterized by high doping levels of heteroatoms N and O.

The above information has been added in the revised manuscript to enrich the description on the chemical environments of these non-metallic elements and the XAFS results have been provided as new Supplementary Fig. 16.

Pages 9-10, Lines 239-258:

“The chemical environments of non-metallic elements (C, N and O) were further investigated by the near-edge X-ray absorption fine structure (NEXAFS) spectroscopy. As shown in the C K-edge NEXAFS spectrum of MCC-950 (Supplementary Fig. 16a), the peaks at 285.5, 286.8 and 287.5 eV are assigned to π^ (C=C), π^* (C-OH or C-O-C) and π^* (N-C=N or C=O) resonances, respectively⁴⁷; while the broad peak centered within 290-295 eV originates from σ^* (C-N) resonance^{47,48}. This indicates the dominance of sp^2 -hybridized conjugated carbon-based framework coupled with the formation of functional groups enabled by heteroatom (C and N) doping. In the N K-edge NEXAFS spectrum, the peaks at 399.8, 401.3 and 407.7 eV may originate from π^* (pyridinic N), π^* (pyrrolic N) and σ^* (C-N) resonances^{48,49}, respectively. The characteristic peak assigned to metal-pyridinic N bonding generally locates between that of pyridinic N and pyrrolic N^{49,50}, which is however not identified in our case (Supplementary Fig. 16b). The observed asymmetrical peak at 401.3 eV may overlap with that of Zn-pyrrolic (not deconvoluted here), which is also supported by the binding energy shift of pyrrolic N in the XPS N 1s spectrum of MCC-950 when compared with other samples (Fig. 3e). The O K-edge spectrum shows a set of peaks at 530-535, 540.4 and 542-545 eV, which are indicative of π^* (C-O-C or O-C=O), σ^* (C-O) and σ^* (C=O) resonances, respectively (Supplementary Fig. 16c)^{47,51}. These results keep good consistency with XPS analyses, which adds further information of the local*

coordination structure of the involved elements in MCC-950, again corroborating that the highly crystalline sp^2 -hybridized carbon network is characterized by high doping levels of heteroatoms N and O.”

Fig. R3 NEXAFS spectra of MCC-950. **a**, C K-edge. **b**, N K-edge. **c**, O K-edge.

The related procedures and fitting methods for all the above characterizations have also been added in “*Materials and methods*” section, which are also listed as follows:

Page 17-18, Lines 492-518:

“C, N and O K-edge NEXAFS spectra were recorded at the photoemission end-station (BL10B) beamline of National Synchrotron Radiation Laboratory

(NSRL) in Hefei, China. Zn K-edge analysis was performed with Si (111) crystal monochromators at the BL11B beamlines at the Shanghai Synchrotron Radiation Facility (SSRF) (Shanghai, China). Before the analysis at the beamline, samples were pressed into thin sheets with 1 cm in diameter and sealed using Kapton tape film. The XAFS spectra were recorded at room temperature using a 4-channel Silicon Drift Detector (SDD) Bruker 5040. Zn K-edge extended X-ray absorption fine structure (EXAFS) spectra were recorded in transmission mode. Negligible changes in the line-shape and peak position of K-edge XANES spectra were observed between two scans taken for a specific sample. The XAFS spectra of these standard samples (ZnPc, ZnO and Zn foil) were also recorded in transmission mode. The acquired EXAFS data were extracted and processed according to the standard procedures using the ATHENA module implemented in the IFEFFIT software packages^[R16]. Subsequently, k^3 -weighted $\chi(k)$ data in the k -space were Fourier transformed to real R space using a hanning windows ($dk = 1.0 \text{ \AA}^{-1}$) to separate the EXAFS contributions from different coordination shells. To obtain the quantitative structural parameters around Zn atoms in the sample, least-squares curve parameter fitting was performed using the ARTEMIS module of IFEFFIT software packages^[R17]. Effective backscattering amplitudes $F(k)$ and phase shifts $\Phi(k)$ of all fitting paths were calculated by the ab initio code FEFF8.0^[R18]. For the sample, a k range of $2.9\text{-}12.45 \text{ \AA}^{-1}$ was used and curve fittings were done in the R -space within the R range of $1\text{-}2 \text{ \AA}$ for k^3 -weighted $\chi(k)$ functions. The number of independent points is:

$$N_{\text{ipt}} = \frac{2 \times \Delta k \times \Delta R}{\pi} = \frac{2 \times (12.45 - 2.9) \times (2 - 1)}{\pi} \approx 6.08$$

For this sample, the Fourier-transformed curves showed two single prominent coordination peaks at 1.5 \AA assigned to the Zn-N/O coordination. During the curve fitting for the sample, Debye-Waller factors (σ^2), coordination numbers (CN), interatomic distances (R) and energy shift (ΔE_0) were treated as

adjustable parameters for the Zn-N path. The number of adjustable parameters was $N_{para} = 1 + 1 + 1 + 1 = 4$, less than the N_{ipt} ."

[R1] Angew. Chem. Int. Ed. 2021, 60, 181-185.

[R2] J. Am. Chem. Soc. 2018, 140, 12469-12475.

[R3] Adv. Mater. 2022, 34, 2110653.

[R4] Proc. Natl. Acad. Sci. USA 2022, 119, e2119492119.

[R5] Appl. Catal. B 2020, 279, 119363.

[R6] Angew. Chem. Int. Ed. 2023, 62, e202219178.

[R7] Angew. Chem. Int. Ed. 2022, 61, e202110838.

[R8] Chem. Eng. J., 2023, 474, 145973.

[R9] Adv. Sci., 2023, 10, 2304088.

[R10] Proc. Nat. Acad. Sci., 2023, 120, e2221228120.

[R11] Angew. Chem. Int. Ed. 2023, 62, e202303525.

[R12] Cell Rep. Phys. Sci. 2020, 1, 100145.

[R13] Nat. Commun. 2019, 10, 1278.

[R14] Angew. Chem. Int. Ed. 2017, 56, 610-614.

[R15] Nat. Catal. 2018, 1, 282-290.

[R16] J. Synchrotron Radiat. 2001, 8, 322-324.

[R17] J. Synchrotron Radiat. 2005, 12, 537-541.

[R18] Phys. Rev. B 1998, 58, 7565-7576.

2. Theoretical calculations are suggested to simulate the proposed atomic structure of crystalline material.

Response: We understand the concern from Reviewer #2 regarding the atomic structure. However, the current technique can barely distinguish the light elements (C, N and O in our case) directly at atomic scale in an unknown material with absolute resolution. We have employed existing structural characterization techniques, such as X-ray Photoelectron Spectroscopy (XPS) and X-ray Absorption Fine Structure (XAFS) spectroscopy, as suggested by

Reviewer #2. These methods provide valuable information about the atomic coordination environment and offer insights into the local structural information of the material.

When it comes to the overall structural information, the analysis of XRD and HR-TEM results in the main text reveals that MCC is characterized by a typical stacking structure in 2D fashion, coming with good crystallinity and long-range order along the c-axis. Our findings indicate that the overall structure of MCC in the c-direction is essentially consistent with that of a nitrate anion-intercalated graphite (graphite intercalation compound), which exhibits AB/BCBC/CACA/AB stacking (as shown in Figure 1)^[R19]. Notably, nitrate anion-intercalated graphite compounds can only exist stably at -35°C, maintaining their intercalation structure at this temperature. However, at room temperature, these intercalation structures collapsed. In contrast, the MCC sample retains a well-defined layered structure for an extended period (> 1 year) at room temperature. This observation suggests that MCC is distinct from nitrate intercalated graphite compounds. When considering the role of Zn species in the sample, we observed that the layered structure collapses upon zinc removal. This indicates that zinc likely acts as an intercalation agent, stabilizing the interlayer structure (as we discussed already in the main text). The structure of MCC along the c-axis has been depicted in Figure 1, providing a clear illustration of its interlayer arrangement. Other peaks observed in the XRD results of MCC, as also depicted in the diffraction findings, are likely associated with the ab-plane structure of MCC, indicating a less pronounced regular structure.

It is noteworthy that the final MCC sample inevitably comprises a mixture of multiple phases, including highly crystalline MCC and amorphous carbon phases as well as graphitic regions. Some of the weaker XRD peaks may originate from these less crystalline phases. These characteristics add substantial difficulty in simulating the fine structure using XRD and HR-TEM data.

[R19] Nature, 1965, 206, 1352-1354.

3. The role of sodium chloride in the carbonization process should be considered. Can crystalline nanocarbon be obtained by directly calcining the m-ZIF-8 in the absence of NaCl? Will NaCl be retained in the obtained material? Is the residual NaCl a part of the composition of the crystalline nanocarbon?

Response: It is evident that NaCl indeed holds significant importance in the carbonization process. We have compiled a comprehensive summary of relevant research and advancements concerning the utilization of NaCl-assisted synthesis for carbonaceous materials in Supplementary Note 1. The functions of NaCl can be broadly categorized into two aspects. First, it serves as a salt reactor, acting as a protective medium during carbonization. This capability helps mitigate the violent decomposition of m-ZIF-8 under high-temperature conditions and aids in preserving the parent microstructure. Here, we also investigated the XRD pattern of the directly carbonized m-ZIF-8 without NaCl addition (Fig. R4, green line), it is clear that this direct carbonization process fails to yield crystalline products, underscoring the supportive role of NaCl as protection agent. Moreover, when m-ZIF-8 was physically mixed with NaCl and subsequently pyrolyzed, the resulting product remains amorphous (Fig. R4, grey line). Therefore, it is clear that the pyrolysis method with absence of salt protection is likely to introduce a substantially larger proportion of amorphous carbon phase, this also holds true for the synthesis by simple mixing of m-ZIF-8 and NaCl. Secondly, NaCl itself serves as a templating agent and facilitates in-situ exfoliation during the carbonization process, thereby fostering the formation of characteristic 2D materials.

We monitor the possible presence of Na in the final product MCC. By investigating multiple-batch samples, we found that the maximum value of Na content detected by inductively coupled plasma optical emission spectroscopy (ICP-OES) in multiple samples is 0.1 wt%, which is quite low. In the coexisting ion interference experiment (please refer to the response to Points #6 below)

that we supplemented, the results also clearly point to the fact that either Na⁺ or Cl⁻ barely affects the catalytic performance. Therefore, we believe that there is no significant NaCl residue in the MCC material, and it is unlikely that Na participates in constructing the primary structure of crystalline nanocarbon or influencing the Fenton-like reaction.

Fig. R4 XRD patterns of m-ZIF-8-C and m-ZIF-NaCl-mix-C.

4. Line 125, page 5. “remains stable in air atmosphere for at least one year”. Is there any experimental data?

Response: We appreciate Reviewer #2 for pointing out this issue. We have supplemented the XRD pattern of the sample after one-year exposure in air atmosphere. It is evident that the XRD pattern remains unchanged when compared to that of fresh one (Fig. R5). This clearly points to the fact that the structure of MCC-950 is well preserved, which coming with long-term stability when exposed in air atmosphere.

We have added this information into the revised main text, the XRD pattern has been added as new Supplementary Fig. 6.

Page 6, Lines 125-127:

“In contrast, MCC-950 can well retain its intercalated structure upon exposure in air atmosphere for at least one year, as reflected by the well preserved XRD pattern (Supplementary Fig. 6).”

Fig. R5 Comparison of the XRD results of fresh MCC and MCC after preservation in environmental atmosphere.

5. Line 153, page 6. It is unreasonable to use thermogravimetry to investigate the thermal stability of materials as phase transformation may also occur at high temperatures.

Response: We agree with Reviewer #2 that phase transformation may occur at elevated temperatures. However, thermogravimetry can reflect the thermal stability of a material to a certain extent, that is, if the material undergoes severe mass loss when raised to a certain temperature, it may indicate violent decomposition. According to the thermogravimetric result of MCC, as the temperature gradually increased around 800°C, no rapid loss of MCC mass was observed. It's noteworthy that our catalytic application operates at room temperature and does not impose stringent requirements on the thermal stability of MCC. That being said, we are aware that this issue should be toned down. Hence, we have refined our description of MCC's thermal stability based on thermogravimetric result to avoid misunderstanding, which is also shown as follows.

Page 7, Lines 154-156:

"The thermal gravimetric analysis (TGA) shows a ~80 wt% residual at 1000 °C, implying the limited mass loss of the MCC-950 at high temperature (Supplementary Fig. 10)."

6. The roles of Zn and residual NaCl in the Fenton reaction should not be neglected.

Response: Yes. As we responded in Point #3, the possible residual of Na in our sample was excluded. In the coexisting ion interference experiment, we assessed the influence of Na⁺ and Cl⁻ on the degradation process, which show that both ions had negligible impact on catalysis (Fig. R6a).

When it comes to the possible catalytic role of zinc, we also performed experiments to afford comprehensive examination. Specifically, we introduced additional 1.5 ppm Zn²⁺ (a concentration consistent with Zn content in the catalyst MCC-950, added in the form of ZnSO₄ solution) as homogeneous species into the catalytic system (Fig. R6b), which can barely affect the catalytic performance in Fenton-like reaction. Considering the potential influence of the different chemical forms of zinc, we also introduced ZnO (again with the same Zn content relative to that of MCC-950) as heterogeneous species into the system, and again, found no variation of the catalytic performance (Fig. R6b). These results indicate that neither Zn²⁺ (heterogeneous) nor Zn(II) (homogeneous) can affect the catalytic process.

As we stated before in Point #1, the fully occupied 3d¹⁰ electronic configuration of zinc inherently disallows the activity in Fenton-like reaction, that is why there have been few reported cases. Efficient catalysts typically do not rely on zinc element as the catalytic center; instead, the Zn element often serves to modulate the electronic structure of other catalytically active centers (refer to Point #1)^[R8-10]. In light of these comprehensive analyses, we can rule out the potential impact of the small amount of Zn remaining in the MCC material on the catalytic process.

The supplemented results have been added and updated as new Supplementary Fig. 23.

Fig. R6 Catalytic degradation profiles in different systems. a, Coexistence of different ions. **b**, Blank systems with coexistence of Zn²⁺ and Zn(II). Reaction condition: [pollutants]=20 mgL⁻¹, [PMS] = 0.4 g L⁻¹, [Zn²⁺/Zn(II)]=0.2 mg, T = 298K.

REVIEWERS' COMMENTS

Reviewer #2 (Remarks to the Author):

The authors have addressed all the concerns and the manuscript is recommended for publication in Nature Communications as is.

Point-by-point responses to reviewers' comments

REVIEWERS' COMMENTS

Reviewer #2 (Remarks to the Author):

The authors have addressed all the concerns and the manuscript is recommended for publication in Nature Communications as is.

Response: The authors appreciate Reviewer #2 for her/his positive feedback on our revision.